# Influencing Factors of Depression among Adolescent Asians in North America: A Systematic Review

**DOI:** 10.3390/healthcare9050537

**Published:** 2021-05-04

**Authors:** Ping Zou, Annisa Siu, Xiyi Wang, Jing Shao, Sunny G. Hallowell, Lihua Lydia Yang, Hui Zhang

**Affiliations:** 1School of Nursing, Nipissing University, Toronto, ON M5T 1V4, Canada; 2Faculty of Health Sciences, McMaster University, Hamilton, ON L8S 4L8, Canada; siua4@mcmaster.ca; 3School of Nursing, Shanghai Jiao Tong University, Shanghai 200025, China; wangxiyi4869@shsmu.edu.cn; 4Faculty of Nursing, Zhejiang University School of Medicine, Hangzhou 310058, China; shaoj@zju.edu.cn; 5College of Nursing, Villanova University, Villanova, PA 19085, USA; sunny.hallowell@villanova.edu; 6Wellness Counselling Centre for Youth Canada, Markham, ON L3R 6G2, Canada; lydia@counsellingcentreforyouth.ca; 7Department of Cardiology, Guizhou Provincial People’s Hospital, Guiyang 550002, China; zhanghui88640@163.com

**Keywords:** influencing factors, depression, adolescent, Asians, North America, systematic review

## Abstract

Background: Asian American adolescents experience rates of depression comparable to or greater than those of other ethnic minorities. The purpose of this systematic review is to summarize psychosocial factors related to depressive symptoms of Asian American adolescents between the ages of 10 to 19. Methods: Various electronic databases were systematically searched to identify research articles published from 2000 to 2021, and the psychosocial factors influencing depression among Asian adolescents in North America were examined. Results: A total of 81 studies were included in this systematic review. Consistent findings on relationships between depressive symptoms and influencing factors included (a) acculturative stress, (b) religious or spiritual significance for females, (c) parent–child cohesion, (d) harsh parenting style, (e) responsive parenting style, (f) racial or ethnic discrimination, (g) being bullied, (h) positive mentor presence, and (i) exposure to community violence. Collectively, the majority of included studies suggest that depressive symptoms were more likely found among Asian American adolescents who (a) are older, (b) are female, (c) have immigrant status, (d) exhibit coping behaviours, (e) face academic challenges, (f) face a poor socioeconomic situation, (g) perceive parent–child conflict, (h) perceive maternal disconnectedness, and (i) perceive negative peer relations. A number of conflictive findings also existed. Discussion/Conclusions: This systematic review provides a summary of the various psychosocial factors on individual, familial, and social levels, which influenced the depressive symptoms of Asian American adolescents. Such findings offer a starting point to examine what factors should be necessarily included in related depression-preventive intervention design and evaluation. Culturally sensitive care, strengthened family–school–community collaboration, and targeted research efforts are needed to meet the needs of Asian adolescents experiencing a range of depressive symptoms.

## 1. Introduction

Depression is defined as a mood disorder that causes persistent feelings of sadness and loss of interest [1]. Symptoms include persistent feelings of hopelessness, sadness, worthlessness, guilt, irritability, lethargy, or “emptiness” and may manifest as physical aches, pains, headaches, cramps, or irregular eating/sleeping habits. Adolescents may further experience an inability to focus and follow through with tasks, likely resulting in lower academic performance or disinterest in usual activities. More severe symptoms include substance abuse, strong feelings of guilt, and panic attacks, with the most serious one being recurrent thoughts of death or suicide [2]. Lifelong consequences can include impaired educational or vocational achievement, impaired relationships, and severe loss of self-esteem [1]. In the United States, undiagnosed depression has resulted in a surge in gun violence and shootings in schools and communities [3]. Within the last decade (specifically, 2011–2016), the prevalence of depression in American adolescents has increased from 8.3% to 12.9%. Rates of depression are even greater among females and older adolescents aged 14 to 19 [4,5]. Existing literature recognizes this upward trend and urges the need for improved screening and access to treatment for adolescents [4,6,7].

Asian Americans represent one of the fastest growing ethnic groups in North America, making up 6.5% of the U.S. population and 17.7% of the Canadian population [8,9]. Asians between the age of 5 to 17 are projected to increase from 5.58% to 8.09% of the resident population in the U.S. by 2060 [10]. In 2016, 27% of Canadian youth aged 15 to 30 belonged to a visible minority group, who are non-Caucasian in race or non-White in colour other than aboriginal peoples, and this number is only projected to increase [11,12]. Thus, it is important to identify influencing factors that are linked to Asian American communities. The perception that Asian Americans disproportionately achieve higher economic and educational success has perpetuated the myth of Asian Americans as the “model minority” and resulted in inadequate assessment and treatment of physical and mental health problems in Asian communities [13]. Furthermore, as of 2003, research found that 47% of Asian American adolescents experience depression compared to 30% of their European American counterparts [14]. This finding is especially concerning when considering the evidence that cultural barriers and biases have masked the true rates of depression among Asian American adolescents [1,15,16]. There is a need for more accurate identification of Asian American adolescents at risk for depression and increased accessibility to mental health services [16,17]. A summary of psychosocial factors may offer insight into the protective factors, screening interventions, and culturally sensitive treatments that should be further investigated.

A key gap in this emergent field is the lack of specificity when addressing Asian American adolescent populations in terms of ethnicity and age. The most comprehensive systematic review of factors in this field included studies published between 2000 and 2015, included Native Hawaiian and Pacific Islander youth, and had a broad age range of 18 or younger [1]. Other reviews encompassed diverse minority groups, including African American and Latino groups, or addressed general outcomes such as well-being [18,19,20]. Since then, research has advanced, and new studies regarding Asian American adolescent depressive symptoms have been published. In response, this present review seeks to synthesize the essential results of studies identifying upstream risk and protective factors focused on depressive symptoms, rather than suicidality or general well-being, in Asian Americans specifically. This review further distinguishes adolescents as youth aged 10 to 19 rather than 18 years old and younger as done before [1]. This is an important consideration since youths aged younger than 10 are in a unique developmental period and may interact differently at individual, familial, and societal levels. Furthermore, the onset of COVID-19 has elevated the risk of violence and racial discrimination against Asian Americans [21,22]. A recent study found that increased perceived racism due to the pandemic is associated with poorer mental health in youth aged 10 to 18 years old [22]. Such findings bring to light many racially and ethnically associated burdens that may have lacked serious investigation previously [21]. Overall, the longstanding issue of disentangling factors influencing the depressive symptoms of Asian Americans aged 10 to 19 has yet to be clearly addressed.

The ecosocial theory is used to guide data collection, data analysis, and presentation of findings in this review. The ecosocial framework construes human development as a process arising from complex interactions between a person and their environment on intrapersonal, interpersonal, and broader societal levels [23,24]. This theory has been used in studies examining factors influencing healthy eating in aged Chinese Canadians, the impact of the financial recession on student mental health, and how discrimination manifests as health inequities [24,25,26]. A recent study addressing various risk factors for depression in U.S. children used a modified ecosocial framework to account for dynamic inferences in a hierarchical structure [27]. The present review applies this theoretical framework to conceptualize the relationship between illness, distribution of depressive symptoms, and the resources or experiences that impact Asian American adolescents [24]. This present review applies the ecosocial framework to summarize existing research on the influencing factors of depression among Asian American adolescents. Our research questions were as follows: (1) What are the individual factors influencing the depressive symptoms of this population? (2) What are the familial factors influencing depression of this population? (3) What are the community and social (cultural/acculturation, racial/ethnic, etc.) factors influencing depression of this population?

## 2. Methods

The protocol and reporting of the results of this systematic review were based on the PRISMA statement [28].

### 2.1. Eligibility Criteria

Studies were included if they satisfied the following criteria: (a) included Asian adolescents who were 10 to 19 years of age; (b) identified a dimension of depression, such as depressive symptoms or depressed mood, as one of the outcome variables; (c) focused on the North American context; (d) examined the influence of psychosocial factors on depressive symptoms; and (e) were published in an English, peer-reviewed journal. Depressive symptoms can be assessed by different instruments, such as the Centre for Epidemiologic Studies Depression Scale [29], the 13-item Short Mood and Feelings Questionnaire [30], the Depression Self-Rating Scale for Children [31], the Beck Depression Inventory—Second Edition [32], and the Children’s Depressive Inventory [33,34,35,36]. Studies were excluded if they met the following conditions: (a) did not have an author; (b) did not have predicted depression status or depressive symptoms as an outcome; (c) did not include Asian American groups in the sample of a study; (d) addressed adolescents under the age of 10 or over the age of 19; (e) focused on interventions or measurement validity and did not address psychosocial factors; (f) were a review, commentary, or dissertation; (g) were on an unrelated topic; (h) had no full text available.

### 2.2. Information Sources

Various health-related, psychological, sociological, and educational science databases, including MEDLINE, PsycINFO, Embase, CINAHL, ProQuest, Nursing and Allied Health Database, PsycARTICLES, and Sociology Database were selected for the literature search.

### 2.3. Search Strategy and Selection of Evidence

The databases were systematically searched using the combination of keywords, (Asia* OR India* OR Afghan* OR Bengal* OR Bangla* OR Bhutan* OR Nepal* OR Pakistan* OR Sri Lanka* OR Cambodia* OR Chin* OR Filipi* OR Taiwan* OR Korea* OR Japan* OR Vietnam* OR Thai*) AND (North America* OR Canad* OR America* OR USA OR U.S.A* OR United State*) AND (adolesc* OR teen* OR youth* OR child* OR young OR pediatric*) AND (depress*). The citations were exported into EndNote to remove any duplicates. The titles and abstracts of all citations were screened for relevance based on the established eligibility criteria. All eligible articles were searched for full-text documents, and full-text documents were carefully reviewed. Further, the reference lists of all eligible articles were manually searched for additional titles not returned in the initial search. The most recent search was conducted in January 2021.

### 2.4. Quality Assessment

The United Kingdom’s Critical Appraisal Skills Programme checklists (https://casp-uk.net/casp-tools-checklists/) are used as quality assessment tools to assess included articles [37]. These checklists are not designed to generate a final quantitative score but to draw attention to elements of a rigorous study and evaluate the study as a whole. Using these checklists, we classified the quality of included papers as low, moderate, or high. Two researchers independently evaluated each article, discussing disagreements in quality ratings according to the guidelines until consensus was reached. Papers rated as low quality were excluded. Thus, all papers included in this review were of moderate to high quality.

### 2.5. Data Extraction

Data were independently extracted by two reviewers based on predetermined criteria. From each article, various data, including authors, year of publication, study population, research design, recruitment method, sample size, sample characteristics, comparison group, outcomes, measurements, and significant findings, were extracted. The data were collected and organized into an Excel spreadsheet. The reviewers discussed disagreements regarding data extraction until consensus was reached.

### 2.6. Data Analysis and Synthesis of Results

Once the data was organized in Excel, descriptive statistics were used to present the characteristics of included studies. Thematic analysis was then used to summarize the findings to each research question. Categorization results were compared among reviewers, and any disagreements among reviewers were resolved with a consensus decision.

## 3. Results

### 3.1. Characteristics of Included Studies

In this review, 81 studies were included (Figure 1). Thirty-seven (45%, 37/81) studies were original, independent studies, while forty-four (54%, 44/81) consisted of secondary analyses of national surveys. Only considering Asian American adolescents and not comparisons or parent groups, the sample size among independent studies ranged from 26 [38] to 451 [39] adolescents. Among studies conducting secondary analysis of national surveys, the sample size ranged from 84 [40] to 1664 [41] adolescents. Only three (3.7%, 3/81) studies were qualitative, and seventy-eight (96.2%, 78/81) were quantitative in methodology. There were two (2.4%, 2/81) studies conducted in Canada, seventy-eight (96.2%, 78/81) in the United States, and one (1.2%, 1/81) in both countries. Regarding participant ethnicity background, thirty-six (44.4%, 36/81) included Chinese adolescents, nineteen (23.4%, 19/81) included Korean, fifteen (18.5%, 15/81) included Vietnamese, eight (9.8%, 8/81) included Filipino, six (7.4%, 6/81) included Japanese, seven (8.6%, 7/81) included Cambodian, three (3.7%, 3/81) included Pakistani, and one (1.2%, 1/81) included Bhutanese adolescents. Three (3.7%, 3/81) studies examined only females, while the remaining seventy-eight (96.2%, 78/81) examined both genders.

### 3.2. Individual Factors

#### 3.2.1. Ascribed Indicators (Age, Gender, Immigration Status, Ethnicity)

Ascribed indicators are social categorizations consisting of inherited or ascribed traits such as age, gender, immigration status, or ethnicity. A total of twenty-eight (34.6%, 28/81) studies addressed the relation between depressive symptoms and at least one ascribed indicator [32,35,36,41,42,43,44,45,46,47,48,49,50,51,52,53,54,55,56,57,58,59,60,61,62,63,64,65]. Six (7.4%, 6/81) studies examined the correlation between age and depressive symptoms [35,42,43,44,49,54]. Of these, four (4.9%, 4/81) studies found that depressive symptoms increased with age [35,42,43,54], while another found depressive symptoms to decrease with age [44]. One study found age to not be associated with depressive symptoms [49]. Thus, the correlation between age and depressive symptoms was inconclusive, with a majority of studies identifying a positive correlation.

Gender was assessed in thirteen (16%, 13/81) studies, twelve (14.8%, 12/81) of which reported that female adolescents experience more numerous or more severe depressive symptoms [35,36,41,45,48,52,53,55,59,64,65,66]. Of the twelve studies, one study further found the relation between perceived chronic daily discrimination and severe depressive symptoms to be significant for females only [53]. The remaining study found no difference in rates of depressive symptoms between male and female adolescents, making our findings inconclusive [62]. Nevertheless, a majority of studies found that female adolescents experienced greater depressive symptoms.

A total of eight (9.8%, 8/81) studies addressed immigrant status [32,36,46,52,58,59,62,64]. Of these, seven (8.6%, 7/81) studies found that being foreign born or being the child of an immigrant was associated with more depressive symptoms [32,46,52,58,59,62,64]. However, one (1.2%, 1/81) contradictory study found that generational status had no significant relation to depression [36]. Findings regarding immigrant status were inconclusive, though a majority found a positive correlation with depressive symptoms.

Nine (11.1%%, 9/81) studies identified a relationship between ethnicity and depressive symptoms [44,55,56,61,62,65,67,68,69]. Of these, six (7.4%, 6/81) found that adolescents of Chinese, Filipino, Vietnamese, or an unidentified Asian ethnicity were more likely to experience depressive symptoms [44,55,56,61,65,69]. The remaining three (3.7%, 3/81) studies found that ethnicity did not predict depressive symptoms [62,67,68]. Thus, the correlation between ethnicity and depressive symptoms was wholly inconclusive.

#### 3.2.2. Acculturation Factors (Acculturation, Generational Status, Coping Mechanism)

Acculturation is defined as the process of individual change and adaptation as a result of continuous contact with new, distinct mainstream culture such as language, identity, values, and behaviours [49]. Acculturative stress refers to the negative psychological impact of this process [57]. A total of thirteen (16%, 13/81) studies addressed the influence of acculturative stress, or related factors such as length of stay and acculturative coping mechanisms, on depressive symptoms [33,38,43,47,49,51,57,61,62,70,71,72,73]. Of these, six (7.4%, 6/81) studies consistently found that acculturative stress was related to greater reporting of depressive symptoms, particularly low self-esteem and somatic symptoms [33,38,43,49,57,62].

Generational status and length of stay are factors relevant to the acculturative process. Three (3.7%, 3/81) studies addressed such factors, one of which found that increased length of stay was non-significantly linked to reduced depressive symptoms [51]. Similarly, another study found that generational status was not related to depressive symptoms [49]. In contrast, the third study found that third-generation Chinese Americans scored higher than other generations in somatic symptoms such as poor appetite, trouble falling asleep, and frequent crying spells [74]. Thus, the relation between increased length of stay and depressive symptoms was inconclusive.

Furthermore, adolescents cope with acculturative stress in different ways, which may also mediate the relationship with depression as addressed in five (6.2%, 5/81) studies. One study found that adolescents were taught to mistrust others and coped by preparing for bias, which was associated with higher levels of depression and lower self-esteem [47]. Adolescents who coped by practicing wishful thinking were also found to experience more depressive symptoms [70]. Anger suppression and increased emotional sensitivity significantly predicted depressive symptoms [71,72]. However, another (1.2%, 1/81) study found that general emotional suppression did not correlate with depressive symptoms [73]. While the overall findings were inconclusive, the majority of studies found that coping behaviours correlated with more depressive symptoms.

#### 3.2.3. Psychological Indicators (Language Use, Academic Challenges, Religion, Diet/BMI)

Psychological indicators are acquired or modifiable traits, such as language, school performance, religion, diet, and body mass index (BMI). A total of seventeen (21%, 17/81) studies examined the relation between at least one psychological indicator and depressive symptoms [30,33,34,42,46,48,53,54,59,65,73,75,76,77,78,79,80]. English language proficiency and use of English at home were associated with a lower risk of depressive symptoms in two (2.5%, 2/81) studies [34,53]. However, another three (3.67%, 3/81) studies found the use of English at home or English proficiency had no relation to depressive symptoms [33,54,75]. Overall, the findings regarding the use of English were inconclusive.

A total of seven (8.4%, 7/81) studies addressed the relation between academic challenges and depressive symptoms [42,48,54,59,65,73,77]. In four (4.9%, 4/81) studies, worse school performance, low achievement motivation, or worry about testing was associated with increased likelihood of depressive symptoms [42,48,59,77]. However, three (3.7%, 3/81) studies found that grade point average and academic stress did not significantly predict depressive symptoms [54,65,73]. Of these, one study highlighted that many Chinese American adolescents were paradoxically high-achieving students who showed greater depressive symptoms [65]. Overall, findings regarding academic challenges were inconclusive, with a majority of studies finding a positive correlation with depressive symptoms.

Religious identity and daily spiritual experience were significant predictors of reduced depressive symptoms in three (3.7%, 3/81) studies [66,76,78]. Of these, two (2.5%, 2/81) specified the relation between higher personal spirituality and fewer depressive symptoms among female adolescents [46,76]. Overall, religious or spiritual significance was related to reduced depressive symptoms for female, Asian American adolescents only.

Three (3.7%, 3/81) studies addressed the relation between BMI or diet and depressive symptoms; two of which had conflicting findings [30,79,80]. One study found that deviation from the average ethnic group BMI was associated with increased depressive symptoms in East and Southeast Asian female adolescents [80]. The other found correlations between BMI and dieting with depressive symptoms to be insignificant for all Asian ethnic groups [30]. A third study uniquely found that high potato and/or carrot intake was related to reduced depressive symptoms in Asian American adolescent students [79]. Overall, the findings were wholly inconclusive.

### 3.3. Family Factors

#### 3.3.1. Living Situation (Household Size, Socioeconomic Status)

The living situation consists of the family size within a household and economic status. Family size was of interest since it is the main form of social support for many Asian adolescents. A total of five (6.2%, 5/81) studies addressed the association of depression with living situations (i.e., household size, economic situation) [32,43,55,62,77]. One study found a negative correlation between living with both parents and symptoms such as suicidal thoughts and distress [32]. In contrast, another study found that increasing family size was not associated with depressive symptoms [55]. Thus, the general findings were wholly inconclusive.

Regarding socioeconomic status, four (4.9%, 4/81) studies addressed its influence on depressive symptoms [43,55,62,77]. One found that socioeconomic status was not significantly correlated with depressive symptoms [62], while the other found it to be a negative predictor of depression [43]. Similarly, adolescent perception of financial constraint was correlated with increased depressive symptoms [55,77]. Overall, studies were inconclusive, though a majority found poor socioeconomic situation to predict depressive symptoms.

#### 3.3.2. Parent–Child Relations (Conflict; Affection/Cohesion; Parenting Style)

Parent–child relations are defined by the nature of parent–child interactions, which include the experience of conflict, affection, and parenting styles. A total of forty-four (53.1%, 43/81) studies examined the effects of the parent–child relationship on adolescent depressive symptoms [33,34,35,38,39,43,45,50,52,59,62,63,64,65,67,68,70,72,73,75,77,81,82,83,84,85,86,87,88,89,90,91,92,93,94,95,96,97,98,99,100,101,102]. Of these, twenty-two (27.2%, 22/81) studies found that parent–child conflict, involving one or both parents, is associated with increased depressive symptoms [35,38,50,63,65,67,70,72,77,81,82,83,85,86,89,90,91,92,93,94,95,99]. Four (4.9%, 4/81) studies specified that conflict within the parent–child dyad may manifest as a sense of alienation; characterized by a lack of familiarity and communication between parent and child [82,83,90,99]. Among these, one explained that a higher level of depression at one time point predicted a stronger sense of alienation at a later time [99]. This is supported by two studies suggesting a stronger sense of alienation is associated with more depressive symptoms in the future [83,99]. In contrast, only one study addressing Vietnamese Americans found that family conflict was not associated with depressive and anxiety symptoms at a later time [73]. Thus, the findings regarding parent–child conflict were inconclusive, with a majority of studies finding conflict to predict increased depressive symptoms.

Sense of cohesion is typically defined as the degree of emotional bonding and connectedness between family members [72]. Twenty-three (28.4%, 23/81) studies also examined parental affect, cohesion, and warmth [33,34,35,39,43,45,59,62,65,72,75,81,84,87,88,89,92,94,95,96,97,98,103]. Increased parental interest in a child’s feelings and ideas, parental warmth, and cohesion were all associated with less adolescent depressive symptoms in seventeen (21%, 17/81) studies [33,34,35,39,43,48,59,62,65,72,75,81,88,89,95,96,103]. Perceived parental warmth was highest in adolescents with strong identification with both their host and ethnic culture, also known as integrated bicultural identification [33]. One study specified that bonding within the mother–child dyad was especially associated with reduced depressive symptoms in Korean American adolescents [104]. Thus, parental warmth and cohesion was related to reduced depressive symptoms.

A total of seven (8.6%, 7/81) studies examined the specific relationship of either the mother– or father–child dyad on adolescent depressive symptoms [39,45,84,85,87,98,104]. Of these, one study found that both parents had identical influences on their child’s depressive symptom levels [39]. Three studies found that bonding with one’s mother was significantly protective, while the father’s negative attributes such as hostility or low paternal warmth impacted depressive symptoms [67,87,104]. In two studies, maternal hostility positively correlated with depressive symptoms, while maternal connectedness negatively correlated with depressive symptoms; neither study addressed the father–adolescent relationship [45,84]. The remaining study found that mother-reported conflict positively correlated with depressive symptoms but again, left father-reported conflict unaddressed [85]. Overall, the findings regarding mother– and father–child relationships were inconclusive, though a majority of studies found that maternal connectedness reduced depressive symptoms.

Parenting style as a part of the family dynamic was examined in nine (11.1%, 9/81) studies [43,57,64,68,87,99,100,101,102]. Of these, four (4.9%, 4/81) found that overbearing family dynamics or an authoritarian parenting style, characterized by demanding behaviour towards adolescents, led to greater reporting of depressive symptoms [64,68,91,101]. However, it is worth noting one of the four studies was specific to refugee adolescents, a unique circumstance [64]. Another four (4.9%, 4/81) studies found that supportive parenting, characterized by warmth and responsiveness to their child led to reduced depressive symptoms [57,87,100], though one of these specified that only concurrent depressive symptoms were affected [39]. One study examined the influence of promoting children’s understanding of ethnic heritage values and racial barriers in preparation for bias in the community and found no change in adolescent depressive symptoms [102]. Overall, a harsh parenting style was related to increased reporting of depressive symptoms, while a warm parenting style was related to reduced depressive symptoms.

#### 3.3.3. Parental Descriptors (Parental Language Use; Parental Psychological Factors)

Parental descriptors are qualities of the parent that are acquired or modifiable over time, such as language, parental education, and psychological factors (e.g., trauma, depression). A total of nine (11.1%, 9/81) studies examined the relationship between parental descriptors and adolescent depressive symptoms [50,52,54,55,58,76,90,97,102]. Three (3.7%, 3/81) studies examined the relation between parental language use and adolescent depressive symptoms [54,76,90]. One study found that a mothers’ Chinese language proficiency was protective against depressive symptoms when the adolescent was also highly proficient [54]. Another study found that parental use of English at home negatively correlated with depressive symptoms in adolescent girls [76]. The remaining study found that adolescents who translated for their parents were more likely to feel alienated from them, which positively correlated with depressive symptoms [90]. Overall, the studies addressed different dimensions of parental language use and were inconclusive.

Six (7.4%, 6/81) studies examined parental psychological factors such as trauma, depression, and education [50,52,55,58,97,102]. Of these, two (2.5%, 2/81) studies found that parental education positively correlated with initial or later adolescent depressive symptoms [50,55]. One of them hypothesized that it may be due to parents expecting more academically of their children, resulting in greater academic stress [55]. In contrast, one (1.2%, 1/81) study found that parental levels of education had no effect on depressive symptoms [97]. Only one (1.2%, 1/81) study specific to parental trauma found that previous maternal traumatic distress did not have a significant impact on adolescent depressive symptoms [58]. Finally, the remaining study found that parents experiencing depressive symptoms were more likely to practice harsh discipline, a lack of reasoning, and low monitoring of adolescent activities. In turn, this led to more depressive symptoms among their adolescents [52]. Due to the variety of psychological factors addressed, findings regarding psychological factors were inconclusive.

### 3.4. Community and Social Factors

#### 3.4.1. Racial/Ethnic Discrimination

Racial and ethnic discrimination is defined as unfair, differential treatment based on race or ethnicity and includes discrimination enacted by peers, adults, and strangers [49]. A total of twelve (14.8%, 12/81) studies addressed the influence of racial and/or ethnic discrimination on adolescents [40,41,42,49,53,56,63,102,105,106,107,108]. All twelve studies found that experienced or perceived discrimination was related to more severe depressive symptoms. One study specified that daily discrimination resulted in increased reporting of depressive symptoms such as lower grade point average, lower self-esteem, and physical complaints [108]. Two studies specified peer discrimination to be positively correlated with more severe depressive symptoms [40,56]. Another two studies specified a negative school climate, characterized by negative peer interactions and teacher-enacted discrimination, to predict greater depressive symptoms [41,63]. Finally, one study examined the impact of microaggressions; brief and regular verbal and behavioural acts that communicate hostile, negative racial insults [107]. This study found that the experience of microaggresions positively correlated with increased depressive symptoms, particularly among female adolescents [107]. Overall, racial or ethnic discrimination was related to greater depressive symptoms.

#### 3.4.2. Peer Relations

Peer relations are defined as the interrelationships between peers and the adolescent’s sense of belonging, support, and integration. Additionally, it includes dimensions of psychological, verbal, and physical bullying experienced between peers. A total of twelve (14.8%, 12/81) studies examined the association of peer relation factors and adolescent depressive symptoms [31,32,35,40,57,79,82,88,96,105,109,110,111]. Five (6.2%, 5/81) of these studies found that positive peer relations, support, or a sense of belonging was related to significantly reduced depressive symptoms and higher self-esteem [31,35,88,96,105,111]. In contrast, one study found that perceived support from peers of a different ethnicity was related to increased reporting of depressive symptoms [32], while another found no correlation between peer support and depressive symptoms [57]. Overall, findings were inconclusive, though a majority of studies found that positive peer relations was related to fewer depressive symptoms.

A total of four (4.9%, 4/81) studies addressed the influence of bullying or unpopularity on adolescent depressive symptoms [79,82,109,110]. Two (2.5%, 2/81) studies found that victims of bullying were more likely to report more severe depressive symptoms [82,110]. These two studies also found contradictory results regarding adolescent bullies and bystanders who observed this behaviour. One found that bullies and bystanders experienced greater depressive symptoms [110], while the other found no correlation [82]. Another study found that unpopularity among same-ethnicity peers was related to greater depressive symptoms, while unpopularity among cross-ethnicity peers showed no association [109]. The remaining study found that lower rates of bullying on school property was related to increased reporting of depressive symptoms [79]. The study went on to discuss the possibility of unobserved parental or peer support variables that may have buffered the relationship between bullying on school property and adolescent depressive symptoms [79]. Overall, being a victim of bullying was related to more severe depressive symptoms. Other dimensions of bullying or unpopularity were inconclusive.

#### 3.4.3. Broader Community Impact

Broader community impact is defined as the impact of community or school members who may range from strangers to mentor figures and who fall outside of the peer relations category. A total of five (6.2%, 5/81) studies examined such factors [76,103,112,113,114]. Of these, four (4.9%, 4/81) found that the presence of a warm and accepting mentor in an adolescent’s life negatively correlated with depressive symptoms [76,103,112,113]. The mentor figure was identified as a teacher [103] and a religious advisor [76]. Uniquely, the fifth study determined an adolescent’s exposure to community violence to be a very strong predictor of depressive symptoms [114]. Overall, a positive mentor was related to less depressive symptoms, while exposure to community violence was related to greater depressive symptoms.

## 4. Discussion

### 4.1. Summary of Findings

Findings of this review revealed that influencing factors of depression among Asian American adolescents range across the individual, familial, and community environment levels. Individual factors included ascribed indicators, acculturation factors, and psychological indicators; familial factors included living situations, parent–child relations, and parental descriptors; community factors included racial or ethnic discrimination, peer relations, and broader community impacts. Some factors were critical in predicting depressive outcomes, though contrary findings have also been identified. In our systematic review, there were consistent findings on relationships between depressive symptoms and influencing factors, including (a) acculturative stress, (b) religious or spiritual significance for females, (c) parent–child cohesion, (d) harsh parenting style, (e) responsive parenting style, (f) racial or ethnic discrimination, (g) being bullied, (h) positive mentor presence, and (i) exposure to community violence. Collectively, the majority of included studies supported that depressive symptoms were more likely to be among Asian American adolescents who (a) are older, (b) are female, (c) have immigrant status, (d) exhibit coping behaviours, (e) face academic challenges, (f) face poor socioeconomic status, (g) perceive parent–child conflict, (h) perceive maternal disconnectedness, and (i) perceive negative peer relations. The association between depression and ethnicity, length of stay, English language proficiency, religious or spiritual significance for males, BMI and diet, household size, variance in mother– or father–child relationships, parental language use, parental psychological factors, and bullying others or observing bullying were less conclusive due to either contradictory findings or a paucity of evidence.

### 4.2. Individual Factors

Our findings indicated that religious or spiritual significance was related to reduced depressive symptoms for Asian American female adolescents. More than ninety observational studies examining the relation between religion and depression in the general population exist, with the majority stating that those who are more religious experience fewer depressive symptoms [115]. Due to the limited evidence within this review, this is an important influencing factor for future interventions among female Asian American adolescents. Future research should examine why religious or spiritual significance might have a differential impact depending on gender.

Our findings indicated that the co-relationship between female Asian American adolescents and depressive symptoms is supported by a majority of included studies but remains inconclusive. The significance of gender as a potential predictor of depression is important, considering the association is also found among adult Asian immigrant populations [116]. Notably, the study addressing adult Asian Americans found that family conflict was especially detrimental for Asian women who strongly identified with their ethnic culture [116]. Extending this to Asian adolescent females, a stronger ethnic identity may be tied to traditional gender roles. Since females may be expected to spend more time within the family context, family conflict can negatively impact females’ mental health [116]. Gendered mechanisms of psychosocial influence on depressive outcomes requires further research attention.

Our findings indicated that ethnicity is wholly inconclusive in predicting depressive symptoms. Similarly, a systematic review of American adolescents found minimal racial or ethnic differences in adolescent depression in regard to national trends [4]. Still, six studies within this review found a variety of Asian ethnicities to predict greater depressive symptoms than their White counterparts or other ethnic minorities. This is consistent with one review’s finding that Asian Americans experience higher rates of suicide and use fewer mental health services than their white counterparts [16]. Overall, these contradictory findings may point to ethnicity and race being correlated with other factors that more directly influence the rates and severity of depressive symptoms in Asian American adolescents. For example, perceived discrimination is more likely to be experienced by Asian ethnic minorities, and in turn, be linked to depressive symptoms.

Our findings indicated that the co-relationship between academic challenges and depressive symptoms is inconclusive but supported by the majority of studies. This influencing factor is particularly important since Asian Americans emphasize academic value within their ethnic cultures [65,117]. Academic performance is innately tied to family-, school-, and peer-related pressures, which may influence adolescents’ psychological adjustment. However, the experience of academic challenges may also vary across different Asian ethnicities. One study examining high school students in China found academic stress to predict depressive symptoms and highlighted the high-achievement/low psychological adjustment paradox [118]. Thus, future research addressing academic challenges in Asian American adolescents should seek to disaggregate data between Asian ethnic subgroups.

### 4.3. Family Factors

Our findings indicated that parent–child cohesion and a warm, responsive parenting style was related to reduced depressive symptoms, while harsh parenting was related to greater depressive symptoms. This finding is consistent with a systematic review examining parenting styles in relation to depressive symptoms and suicidal ideation in a global population of adolescents [119]. Future studies should focus on the parent–child relationship and parenting style as a point of intervention.

Our findings indicated that the co-relationship between parent–child conflict and depressive symptoms is inconclusive but supported by the majority of studies. This finding is consistent with studies regarding familial conflict and African American, European American, and mainland Chinese adolescent depressive symptoms [117,120,121]. While Asian Americans are comparable to other ethnicities in relation to family factors, they have the unique mediator of acculturative stress, which is thoroughly addressed in the current literature [122]. Since parental support is a critical factor in adolescent development, future research may consider parent–child acculturative discrepancies to be the focus of potential interventions [123]. Furthermore, future research may consider what parent-related facilitators and barriers Asian American adolescents face regarding mental health service utilization due to acculturative discrepancies.

Our findings indicated that the variance in influence of mother– and father–child relationships is inconclusive, though a majority of included studies found maternal connectedness to reduce depressive symptoms. This finding may be due in part to fathers being underrepresented in family research. Future research should address the barriers fathers face and consider how to better engage them in research.

### 4.4. Community and Social Factors

Our findings indicated that racial or ethnic discrimination was related to greater depressive symptoms. This finding is supported by two reviews examining a general sample of adolescents [124] and a sample of minority adolescents in the U.S. [125]. The latter systematic review stressed the importance of applying an intersectional lens when assessing experienced discrimination. This lens considers the cumulative psychological impact of multiple social identities such as ethnicity, class, and gender, as well as individual experiences of discrimination [125]. Future research should seek to apply an intersectional lens to identify high-risk subgroups who are particularly vulnerable to discrimination. This would allow the creation of targeted prevention and treatment interventions.

Our findings indicated that being a victim of bullying and exposure to community violence was related to greater depressive symptoms. Bullying victimization is an important influencing factor due to its volatile impact on adolescents. A systematic review addressing a general sample of adolescents found that depressive symptoms were a mediator for self-harm and school bullying [126]. Furthermore, bullying coupled with mental health disturbances has been found to result in high-risk behaviours among Asian American youth, including heavy drinking, sexual promiscuity, and tobacco use [127]. A review of adolescents in the United States found that exposure to community violence is increasing at exceedingly high rates and is placing adolescents at a higher risk for emotional and behavioural problems such as antisocial and suicidal behaviour [128]. School-based interventions for bullying must be urgently assessed for effectiveness to prevent self-harm, high-risk behaviours, and worsening symptoms of depression.

### 4.5. Integrative View

Among the included studies, a total of forty-two (52.8%, 42/81) studies addressed individual factors, forty-nine (60.5%, 49/81) addressed familial factors, and twenty-six (32.1%, 26/81) addressed community or societal factors. The categorization of these factors through the ecosocial framework allows for simplified conceptualization of the ever-changing relationships between adolescents and their environment, other people, and collective institutions. While classified separately, all interact and influence one another to varying degrees and at different times. Overall, this review presents a need to further investigate community or societal factors to the same depth achieved with individual and familial factors. Considering the conclusive findings regarding community factors (i.e., racial or ethnic discrimination, being bullied, positive mentor presence, and exposure to community violence), this level of social ecology may have imminent implications. The economic, physical, and psychological burdens of COVID-19 on Asian American communities, as well as the rise in violence against Asian Americans, make this need urgent [129]. These broad social burdens may create or severely exacerbate negative individual or familial dynamics. For example, a recent study found that parent’s depressive symptoms worsened with perceived racial discrimination due to COVID-19 [129]. Due to a paucity of evidence, this present review found parental psychological factors to be wholly inconclusive. However, a preliminary study identified parental depression to correlate with harsher parenting practices, which led to worse adolescent depressive symptoms [52]. These are the overlapping complex interactions that may occur, and they require careful consideration. Though COVID-19 highlights such issues, the media coverage of anti-Asian sentiments will inevitably subside. When that occurs, the inequalities and long-lasting burdens faced by Asian American communities will still exist. Clinicians ought to persist in examining how the risk and protective factors identified in this review fit into the unique context of each adolescents’ life.

### 4.6. Limitations

Firstly, this review examined data in aggregate without separating Asian ethnic subgroups, despite Asian Americans being a heterogeneous population. This was due to our research goal of summarizing information focused on a general population of Asian American adolescents. Additionally, subgroup comparisons were not possible across all studies as diverse Asian subsamples were not large enough. The failure to identify individual Asian subgroups could mask disparities and result in inaccurate conclusions regarding assessment, needs, intervention design, and research [130]. Thus, future reviews may consider only including studies with large-enough Asian subsamples for disaggregated data analysis. Secondly, the included papers were heterogenous in their measures of depression, which led to difficulties in comparisons and limited the generalizability of the findings. Assessment of the comparative validity between various measures and methodologies may allow future reviews to overcome this limitation. Thirdly, to reduce risk of bias, included studies were appraised using tools from the Critical Appraisal Skills Programme as done in other systematic reviews [20]. In addition, two independent reviewers screened studies for eligibility, with a third available in the event of disagreements [131]. Future reviews may choose alternative appraisal tools if deemed more methodologically rigorous. Lastly, all studies included within this review were co-relational studies, which does not imply any causal relationship. Relationships discussed in this review require further experimental research to determine causality.

### 4.7. Implications and Future Direction

Findings of this review have implications for healthcare providers, families, and schools. Culturally sensitive care that considers the unique factors influencing depression among Asian American adolescents is crucial. Healthcare providers should take into consideration the central value of parent–child relations in Asian American adolescent lives. Therapeutic conversations with adolescents and their families can offer insight into the adolescents’ functioning, school performance, the influence parents hold, and potential familial barriers in receiving mental health services. This interaction between healthcare providers and adolescents would also provide an opportunity to connect families with more ethnic social support to address potential financial, social, or language burdens. Since parental support is a critical factor in adolescent development, parent–child cohesion and warm, responsive parenting should be promoted. A parent–child conflict should be discussed and solved in a mutually respectful approach. School-based interventions should be encouraged and assessed for their effectiveness on adolescents’ mental health.

This review identified several implications for further research. Firstly, there is a need to disentangle conflicting findings and highlight potential protective factors that are embedded within the results of the literature. For example, future studies should explore potential early life predictors of depression such as age, length of stay, immigrant status, English language use, and socioeconomic status. Secondly, since acculturative discrepancies or parent–child conflict may hinder access to mental health services, future research should examine the cultural stigma surrounding mental health that is present in many Asian American adolescents’ lives [1]. Thirdly, considering the essential role of family relations in Asian American adolescents’ lives, future research should assess the therapeutic effects of family-centred interventions and preventative strategies.

## 5. Conclusions

This systematic review provides a summary of the various psychosocial factors on individual, familial, and social levels, which influence the depressive symptoms of Asian American adolescents. Such findings offer a starting point to examine what factors should be necessarily included in depression-preventive intervention design and evaluation. Culturally sensitive care, strengthened family–school–community collaboration, and targeted research efforts are needed to meet the needs of Asian adolescents experiencing a range of depressive symptoms.

## Figures and Tables

**Figure 1 healthcare-09-00537-f001:**
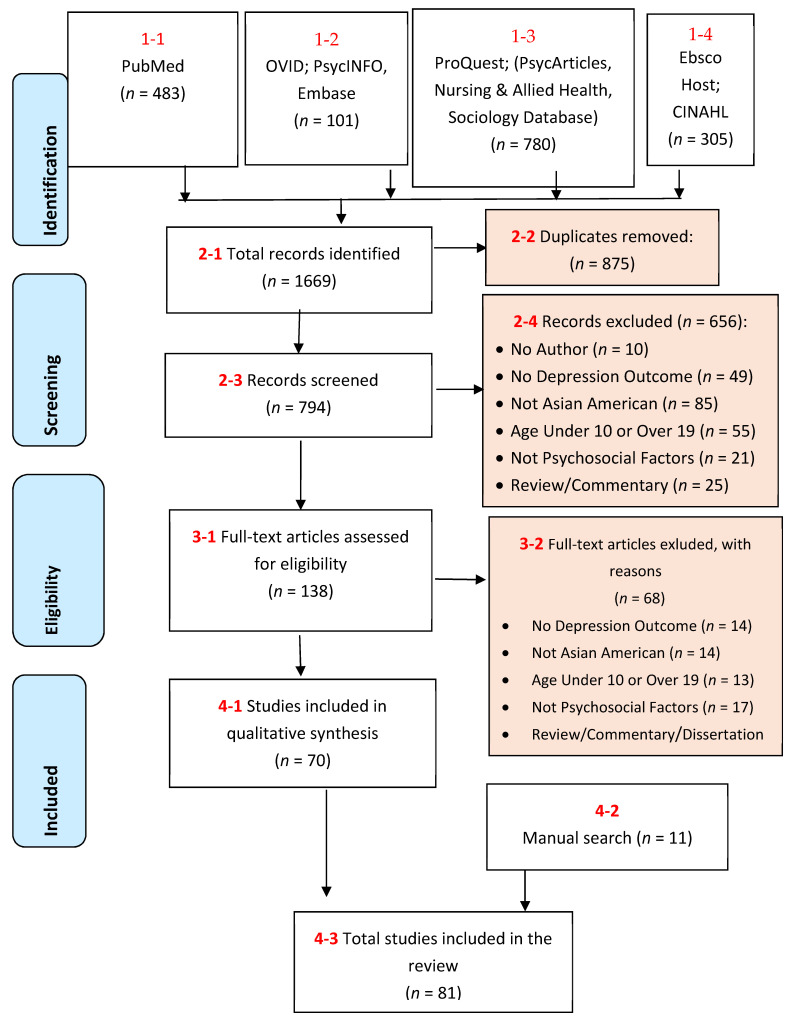
Study selection flow diagram.

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
