# Peer review of "Influencing Factors of Depression among Adolescent Asians in North America: A Systematic Review"

_healthcare, 2021, doi:10.3390/healthcare9050537_

Round 1

Reviewer 1 Report

Thank you for the opportunity to review this paper. Here are some suggestions:

Introduction

  1. Line 58- what does it mean “a visible minority group” ?
  2. Line 71- I agree with the choice to anchor the review in the ecosocial theory. However, I believe that this part of the introduction should be elaborated and include the basic assumption of the theory and its unique contribution to understanding minority stress.
  3. The introduction is well written and clearly presents the research background and questions. It is advised to clarify the rationale of the literature review. In other words, why should Asian adolescents be considered at high risk for depression? Or why should we focus on risk factors for depression specifically in this minority group?
  4. I am wondering about the decision to refer to the whole group of Asian countries as a sole group of “Asian minority”. Except for the fact that all adolescents are originated in the same continent, there seem to be enormous differences in the social and cultural aspects between these countries. The authors may want to refer to this point by defining the term “Asian minority” already in the introduction and address some of these concerns.
  5. I was also wondering what does this specific systematic review contribute to the literature beyond what was also described by other reviews, such as

Wyatt, L.C., et al., Risk Factors of Suicide and Depression among Asian American, Native Hawaiian, 566 and Pacific Islander Youth: A Systematic Literature Review. J Health Care Poor Underserved, 2015. 567 26(2 Suppl): p. 191-237.

Method

  1. Were only quantitative studies included in the review?

Results

  1. Line 188- the term “other ethnic groups” is not clear enough. Please specify.
  2. Line 334- I am not sure that parental education should be considered as a parental psychological factor (more like a demographic one).
  3. Is it correct to place religious or spiritual significance under “individual factors”? Why not cultural factors?
  4. This question is also relevant for the “ethnicity” factor.
  5. Did you find any evidence for the effects of age ?

Discussion

  1. The discussion does a good job in summarizing the results. However, I missed a more integrative view of the above findings and an overall perspective. In other words, except for identifying a wide range of risk factors, what could we learn about this population? It might be useful to integrate these findings with the eco social theoretic framework.
  2. In the limitation section, the authors may want to refer to the risk of bias.

Author Response

Thank you very much for your great feedback. Please see the attachment for our response. 

Reviewer 2 Report

The work is well structured and the review is correct.

I have some suggestions for the authors in order to improve the manuscript, which I explain below.

The introduction only very briefly explains the aim of the study, the research problem and the questions. The topic's relevance is missing, including the study's framing into an important topic and missing "state of the art" or "Literature review". This session should explain the existing research on the study topic and explain how this planning focus is new and stimulating.

In the introduction, the authors should succinctly state the longstanding problem of importance that they seek to clarify through this review.

Author Response

(The authors gave the same response as above.)

Round 2

Reviewer 2 Report

The authors have addressed all my comments, and so, the recommendation for acceptance can be given.